# Consistent Multilabel Classification

**Oluwasanmi Koyejo**[*]
Department of Psychology,
Stanford University
sanmi@stanford.edu

**Nagarajan Natarajan**[*]
Department of Computer Science,
University of Texas at Austin
naga86@cs.utexas.edu

**Pradeep Ravikumar**
Department of Computer Science,
University of Texas at Austin
pradeepr@cs.utexas.edu

**Inderjit S. Dhillon**
Department of Computer Science,
University of Texas at Austin
inderjit@cs.utexas.edu

## Abstract

Multilabel classification is rapidly developing as an important aspect of modern predictive modeling, motivating study of its theoretical aspects. To this end, we propose a framework for constructing and analyzing multilabel classification metrics which reveals novel results on a parametric form for population optimal classifiers, and additional insight into the role of label correlations. In particular, we show that for multilabel metrics constructed as instance-, micro- and macro-averages, the population optimal classifier can be decomposed into binary classifiers based on the marginal instance-conditional distribution of each label, with a weak association between labels via the threshold. Thus, our analysis extends the state of the art from a few known multilabel classification metrics such as Hamming loss, to a general framework applicable to many of the classification metrics in common use. Based on the population-optimal classifier, we propose a computationally efficient and general-purpose plug-in classification algorithm, and prove its consistency with respect to the metric of interest. Empirical results on synthetic and benchmark datasets are supportive of our theoretical findings.

## 1 Introduction

Modern classification problems often involve the prediction of multiple labels simultaneously associated with a single instance e.g. image tagging by predicting multiple objects in an image. The growing importance of multilabel classification has motivated the development of several scalable algorithms [8, 12, 18] and has led to the recent surge in theoretical analysis [1, 3, 7, 16] which helps guide and understand practical advances. While recent results have advanced our knowledge of optimal population classifiers and consistent learning algorithms for particular metrics such as the Hamming loss and multilabel $F$-measure [3, 4, 5], a general understanding of learning with respect to multilabel classification metrics has remained an open problem. This is in contrast to the more traditional settings of binary and multiclass classification where several recently established results have led to a rich understanding of optimal and consistent classification [9, 10, 11]. This manuscript constitutes a step towards establishing results for multilabel classification at the level of generality currently enjoyed only in these traditional settings.

Towards a generalized analysis, we propose a framework for multilabel sample performance metrics and their corresponding population extensions. A classification *metric* is constructed to measure the utility[1] of a classifier, as defined by the practitioner or end-user. The utility may be measured using

---

[*]Equal contribution.

[1]Equivalently, we may define the loss as the negative utility.

the *sample* metric given a finite dataset, and further generalized to the *population* metric with respect to a given data distribution (i.e. with respect to infinite samples). Two distinct approaches have been proposed for studying the population performance of classifier in the classical settings of binary and multiclass classification, described by Ye et al. [17] as decision theoretic analysis (DTA) and empirical utility maximization (EUM). DTA population utilities measure the expected performance of a classifier on a fixed-size test set, while EUM population utilities are directly defined as a function of the population confusion matrix. However, state-of-the-art analysis of multilabel classification has so-far lacked such a distinction. The proposed framework defines both EUM and DTA multilabel population utility as generalizations of the aforementioned classic definitions. Using this framework, we observe that existing work on multilabel classification [1, 3, 7, 16] have exclusively focused on optimizing the DTA utility of (specific) multilabel metrics.

Averaging of binary classification metrics remains one of the most widely used approaches for defining multilabel metrics. Given a binary label representation, such metrics are constructed via averaging with respect to labels (instance-averaging), with respect to examples separately for each label (macro-averaging), or with respect to both labels and examples (micro-averaging). We consider a large sub-family of such metrics where the underlying binary metric can be constructed as a fraction of linear combinations of true positives, false positives, false negatives and true negatives [9]. Examples in this family include the ubiquitous Hamming loss, the averaged precision, the multilabel averaged $F$-measure, and the averaged Jaccard measure, among others. Our key result is that a Bayes optimal multilabel classifier for such metrics can be explicitly characterized in a simple form – the optimal classifier thresholds the label-wise conditional probability marginals, and the label dependence in the underlying distribution is relevant to the optimal classifier only through the threshold parameter. Further, the threshold is *shared* by all the labels when the metric is instance-averaged or micro-averaged. This result is surprising and, to our knowledge, a first result to be shown at this level of generality for multilabel classification. The result also sheds additional insight into the role of label correlations in multilabel classification – answering prior conjectures by Dembczyński et al. [3] and others.

We provide a plug-in estimation based algorithm that is efficient as well as theoretically consistent, i.e. the true utility of the empirical estimator approaches the optimal (EUM) utility of the Bayes classifier (Section 4). We also present experimental evaluation on synthetic and real-world benchmark multilabel datasets comparing different estimation algorithms (Section 5) for representative multilabel performance metrics selected from the studied family. The results observed in practice are supportive of what the theory predicts.

## 1.1 Related Work

We briefly highlight closely related theoretical results in the multilabel learning literature. Gao and Zhou [7] consider the consistency of multilabel learning with respect to DTA utility, with a focus on two specific losses – Hamming and rank loss (the corresponding measures are defined in Section 2). Surrogate losses are devised which result in consistent learning with respect to these metrics. In contrast, we propose a plug-in estimation based algorithm which directly estimates the Bayes optimal, without going through surrogate losses. Dembczynski et al. [2] analyze the DTA population optimal classifier for the multilabel rank loss, showing that the Bayes optimal is independent of label correlations in the unweighted case, and construct certain weighted univariate losses which are DTA consistent surrogates in the more general weighted case. Perhaps the work most closely related to ours is by Dembczynski et al. [4] who propose a novel DTA consistent plug-in rule estimation based algorithm for multilabel $F$-measure. Cheng et al. [1] consider optimizing popular losses in multilabel learning such as Hamming, rank and subset 0/1 loss (which is the multilabel analog of the classical 0-1 loss). They propose a probabilistic version of classifier chains (first introduced by Read et al. [13]) for estimating the Bayes optimal with respect to subset 0/1 loss, though without rigorous theoretical justification.

## 2 A Framework for Multilabel Classification Metrics

Consider multilabel classification with $M$ labels, where each instance is denoted by $x \in \mathcal{X}$. For convenience, we will focus on the common binary encoding, where the labels are represented by a vector $\mathbf{y} \in \mathcal{Y} = \{0, 1\}^M$, so $y_m = 1$ iff the $m^{th}$ label is associated with the instance, and

$y_m = 0$ otherwise. The goal is to learn a multilabel classifier $\mathbf{f} : \mathcal{X} \mapsto \mathcal{Y}$ that optimizes a certain performance metric with respect to $\mathbb{P}$ – a fixed data generating distribution over the domain $\mathcal{X} \times \mathcal{Y}$, using a training set of instance-label pairs $(x^{(n)}, \mathbf{y}^{(n)})$, $n = 1, 2, \ldots, N$ drawn (typically assumed iid.) from $\mathbb{P}$. Let $X$ and $Y$ denote the random variables for instances and labels respectively, and let $\Psi$ denote the performance (utility) metric of interest.

Most classification metrics can be represented as functions of the entries of the confusion matrix. In case of binary classification, the confusion matrix is specified by four numbers, i.e., true positives, true negatives, false positives and false negatives. Similarly, we construct the following primitives for multilabel classification:

$$\widehat{\mathrm{TP}}(\mathbf{f})_{m,n} = [\![f_m(x^{(n)}) = 1, y_m^{(n)} = 1]\!] \qquad \widehat{\mathrm{TN}}(\mathbf{f})_{m,n} = [\![f_m(x^{(n)}) = 0, y_m^{(n)} = 0]\!]$$
$$\widehat{\mathrm{FP}}(\mathbf{f})_{m,n} = [\![f_m(x^{(n)}) = 1, y_m^{(n)} = 0]\!] \qquad \widehat{\mathrm{FN}}(\mathbf{f})_{m,n} = [\![f_m(x^{(n)}) = 0, y_m^{(n)} = 1]\!] \qquad (1)$$

where $[\![Z]\!]$ denotes the indicator function that is 1 if the predicate $Z$ is true or 0 otherwise. It is clear that most multilabel classification metrics considered in the literature can be written as a function of the $MN$ primitives defined in (1).

In the following, we consider a construction which is of sufficient generality to capture all multilabel metrics in common use. Let $A_k(\mathbf{f}) : \{\widehat{\mathrm{TP}}(\mathbf{f})_{m,n}, \widehat{\mathrm{FP}}(\mathbf{f})_{m,n}, \widehat{\mathrm{TN}}(\mathbf{f})_{m,n}, \widehat{\mathrm{FN}}(\mathbf{f})_{m,n}\}_{m=1,n=1}^{M,N} \mapsto \mathbb{R}$, $k = 1, 2, \ldots, K$ represent a set of $K$ functions. Consider *sample* multilabel metrics constructed as functions: $\Psi : \{A_k(\mathbf{f})\}_{k=1}^{K} \mapsto [0, \infty)$. We note that the metric need *not* decompose over individual instances. Equipped with this definition of a sample performance metric $\Psi$, consider the population *utility* of a multilabel classifier $\mathbf{f}$ defined as:

$$\mathcal{U}(\mathbf{f}; \Psi, \mathbb{P}) = \Psi(\{\mathrm{E}\,[\,A_k(\mathbf{f})\,]\}_{k=1}^{K}), \qquad (2)$$

where the expectation is over iid draws from the joint distribution $\mathbb{P}$. Note that this can be seen as a multilabel generalization of the so-called Empirical Utility Maximization (EUM) style classifiers studied in binary [9, 10] and multiclass [11] settings.

Our goal is to learn a multilabel classifier that maximizes $\mathcal{U}(\mathbf{f}; \Psi, \mathbb{P})$ for general performance metrics $\Psi$. Define the (Bayes) optimal multilabel classifier as:

$$\mathbf{f}_{\Psi}^* = \operatorname*{argmax}_{\mathbf{f}:\mathcal{X}\to\{0,1\}^M} \mathcal{U}(\mathbf{f}; \Psi, \mathbb{P}). \qquad (3)$$

Let $\mathcal{U}(\mathbf{f}_{\Psi}^*; \Psi, \mathbb{P}) = \mathcal{U}_{\Psi}^*$. We say that $\hat{\mathbf{f}}_{\Psi}$ is a consistent estimator of $\mathbf{f}_{\Psi}^*$ if $\mathcal{U}(\hat{\mathbf{f}}; \Psi, \mathbb{P}) \xrightarrow{p} \mathcal{U}_{\Psi}^*$.

**Examples.** The averaged accuracy (1 - Hamming loss) used in multilabel classification corresponds to simply choosing: $A_1(\mathbf{f}) = \frac{1}{MN}\sum_{m=1}^{M}\sum_{n=1}^{N}\widehat{\mathrm{FP}}(\mathbf{f})_{m,n} + \widehat{\mathrm{FN}}(\mathbf{f})_{m,n}$ and $\Psi_{\mathrm{Ham}}(\mathbf{f}) = 1 - A_1(\mathbf{f})$. The measure corresponding to rank loss[2] can be obtained by choosing $A_k(\mathbf{f}) = \frac{1}{M^2}\sum_{m_1=1}^{M}\sum_{m_2=1}^{M}\left(\widehat{\mathrm{FP}}(\mathbf{f})_{m_1,k}\right)\left(\widehat{\mathrm{FN}}(\mathbf{f})_{m_2,k}\right)$, for $k = 1, 2, \ldots, N$ and $\Psi_{\mathrm{Rank}} = 1 - \frac{1}{N}\sum_{k=1}^{N}A_k(\mathbf{f})$. Note that the choice of $\{A_k\}$, and therefore $\Psi$, is not unique.

**Remark 1.** *Existing results on multilabel classification have focused on decision-theoretic analysis (DTA) style classifiers, where the utility is defined as:*

$$\mathcal{U}_{\mathrm{DTA}}(\mathbf{f}; \Psi, \mathbb{P}) = \mathrm{E}\left[\Psi(\{A_k(\mathbf{f})\}_{k=1}^{K})\right], \qquad (4)$$

*and the expectation is over iid samples from $\mathbb{P}$. Furthermore, there are no theoretical results for consistency with respect to general performance metrics $\Psi$ in this setting (See Appendix B.2).*

For the remainder of this manuscript, we refer to $\mathcal{U}(\mathbf{f}; \mathbb{P})$ as the utility defined in (2). We will also drop the argument $\mathbf{f}$ (e.g. write $\widehat{\mathrm{TP}}(\mathbf{f})$ as $\widehat{\mathrm{TP}}$) when it is clear from the context.

## 2.1 A Framework for Averaged Binary Multilabel Classification Metrics

The most popular class of multilabel performance metrics consists of averaged binary performance metrics, that correspond to particular settings of $\{A_k(\mathbf{f})\}$ using certain averages as described in the following. For the remainder of this subsection, the metric $\Psi : [0, 1]^4 \to [0, \infty)$ will refer to a binary classification metric as is typically applied to a binary confusion matrix.

**Micro-averaging:** Micro-averaged multilabel performance metrics $\Psi_{\text{micro}}$ are defined by averaging over both labels and examples. Let:

$$\widehat{\text{TP}}(\mathbf{f}) = \frac{1}{MN} \sum_{n=1}^{N} \sum_{m=1}^{M} \widehat{\text{TP}}(\mathbf{f})_{m,n}, \qquad \widehat{\text{FP}}(\mathbf{f}) = \frac{1}{MN} \sum_{n=1}^{N} \sum_{m=1}^{M} \widehat{\text{FP}}(\mathbf{f})_{m,n}, \qquad (5)$$

$\widehat{\text{TN}}(\mathbf{f})$ and $\widehat{\text{FN}}(\mathbf{f})$ are defined similarly, then the micro-averaged multilabel performance metrics are given by:

$$\Psi_{\text{micro}}(\{A_k(\mathbf{f})\}_{k=1}^{K}) := \Psi(\widehat{\text{TP}}, \widehat{\text{FP}}, \widehat{\text{TN}}, \widehat{\text{FN}}). \qquad (6)$$

Thus, for micro-averaging, one applies a binary performance metric to the confusion matrix defined by the averaged quantities described in (5).

**Macro-averaging:** The metric $\Psi_{\text{macro}}$ measures average classification performance across labels. Define the averaged measures:

$$\widehat{\text{TP}}_m(\mathbf{f}) = \frac{1}{N} \sum_{n=1}^{N} \widehat{\text{TP}}(\mathbf{f})_{m,n}, \qquad \widehat{\text{FP}}_m(\mathbf{f}) = \frac{1}{N} \sum_{n=1}^{N} \widehat{\text{FP}}(\mathbf{f})_{m,n},$$

$\widehat{\text{TN}}_m(\mathbf{f})$ and $\widehat{\text{FN}}_m(\mathbf{f})$ are defined similarly. The macro-averaged performance metric is given by:

$$\Psi_{\text{macro}}(\{A_k(\mathbf{f})\}_{k=1}^{K}) := \frac{1}{M} \sum_{m=1}^{M} \Psi(\widehat{\text{TP}}_m, \widehat{\text{FP}}_m, \widehat{\text{TN}}_m, \widehat{\text{FN}}_m). \qquad (7)$$

**Instance-averaging:** The metric $\Psi_{\text{instance}}$ measures the average classification performance across examples. Define the averaged measures:

$$\widehat{\text{TP}}_n(\mathbf{f}) = \frac{1}{M} \sum_{m=1}^{M} \widehat{\text{TP}}(\mathbf{f})_{m,n}, \qquad \widehat{\text{FP}}_n(\mathbf{f}) = \frac{1}{M} \sum_{m=1}^{M} \widehat{\text{FP}}(\mathbf{f})_{m,n},$$

$\widehat{\text{TN}}_n(\mathbf{f})$ and $\widehat{\text{FN}}_n(\mathbf{f})$ are defined similarly. The instance-averaged performance metric is given by:

$$\Psi_{\text{instance}}(\{A_k(\mathbf{f})\}_{k=1}^{K}) := \frac{1}{N} \sum_{n=1}^{N} \Psi(\widehat{\text{TP}}_n, \widehat{\text{FP}}_n, \widehat{\text{TN}}_n, \widehat{\text{FN}}_n). \qquad (8)$$

## 3 Characterizing the Bayes Optimal Classifier for Multilabel Metrics

We now characterize the optimal multilabel classifier for the large family of metrics outlined in Section 2.1 ($\Psi_{\text{micro}}$, $\Psi_{\text{macro}}$ and $\Psi_{\text{instance}}$) with respect to the EUM utility. We begin by observing that while micro-averaging and instance-averaging seem quite different when viewed as sample averages, they are in fact equivalent at the population level. Thus, we need only focus on $\Psi_{\text{micro}}$ to characterize $\Psi_{\text{instance}}$ as well.

**Proposition 1.** *For a given binary classification metric $\Psi$, consider the averaged multilabel metrics $\Psi_{\text{micro}}$ defined in (6) and $\Psi_{\text{instance}}$ defined in (8). For any $\mathbf{f}$, $\mathcal{U}(\mathbf{f}; \Psi_{\text{micro}}, \mathbb{P}) \equiv \mathcal{U}(\mathbf{f}; \Psi_{\text{instance}}, \mathbb{P})$. In particular, $\mathbf{f}^*_{\Psi^*_{\text{micro}}} \equiv \mathbf{f}^*_{\Psi^*_{\text{instance}}}$.*

We further restrict our study to metrics $\Psi$ selected from the linear-fractional metric family, recently studied in the context of binary classification [9]. Any $\Psi$ in this family can be written as:

$$\Psi(\widehat{\text{TP}}, \widehat{\text{FP}}, \widehat{\text{FN}}, \widehat{\text{TN}}) = \frac{a_0 + a_{11}\widehat{\text{TP}} + a_{10}\widehat{\text{FP}} + a_{01}\widehat{\text{FN}} + a_{00}\widehat{\text{TN}}}{b_0 + b_{11}\widehat{\text{TP}} + b_{10}\widehat{\text{FP}} + b_{01}\widehat{\text{FN}} + b_{00}\widehat{\text{TN}}},$$

where $a_0, b_0, a_{ij}, b_{ij}, i, j \in \{0, 1\}$ are fixed, and $\widehat{\text{TP}}, \widehat{\text{FP}}, \widehat{\text{FN}}, \widehat{\text{TN}}$ are defined as in Section 2.1. Many popular multilabel metrics can be derived using linear-fractional $\Psi$. Some examples include[3]:

$$
F_\beta: \quad \Psi_{F_\beta} = \frac{(1+\beta^2)\widehat{\text{TP}}}{(1+\beta^2)\widehat{\text{TP}} + \beta^2\widehat{\text{FN}} + \widehat{\text{FP}}} \qquad \text{Jaccard}: \quad \Psi_{\text{Jacc}} = \frac{\widehat{\text{TP}}}{\widehat{\text{TP}} + \widehat{\text{FP}} + \widehat{\text{FN}}}
$$

$$
\text{Hamming}: \quad \Psi_{\text{Ham}} = \widehat{\text{TP}} + \widehat{\text{TN}} \qquad\qquad \text{Precision}: \quad \Psi_{\text{Prec}} = \frac{\widehat{\text{TP}}}{\widehat{\text{TP}} + \widehat{\text{FP}}}
$$

$$(9)$$

Define the population quantities: $\pi = \sum_{m=1}^{M} \mathbb{P}(Y_m = 1)$ and $\gamma(\mathbf{f}) = \sum_{m=1}^{M} \mathbb{P}(f_m(x) = 1)$. Let $\mathrm{TP}(\mathbf{f}) = \mathrm{E}\left[\widehat{\mathrm{TP}}(\mathbf{f})\right]$, where the expectation is over iid draws from $\mathbb{P}$. From (5), it follows that, $\mathrm{FP}(\mathbf{f}) := \mathrm{E}\left[\widehat{\mathrm{FP}}(\mathbf{f})\right] = \gamma(\mathbf{f}) - \mathrm{TP}(\mathbf{f})$, $\mathrm{TN}(\mathbf{f}) = 1 - \pi - \gamma(\mathbf{f}) + \mathrm{TP}(\mathbf{f})$ and $\mathrm{FN}(\mathbf{f}) = \gamma(\mathbf{f}) - \mathrm{TP}(\mathbf{f})$.

Now, the population utility (2) corresponding to $\Psi_{\mathrm{micro}}$ can be written succinctly as:

$$\mathcal{U}(\mathbf{f}; \Psi_{\mathrm{micro}}, \mathbb{P}) = \Psi(\mathrm{TP}(\mathbf{f}), \mathrm{FP}(\mathbf{f}), \mathrm{FN}(\mathbf{f}), \mathrm{TN}(\mathbf{f})) = \frac{c_0 + c_1 \mathrm{TP}(\mathbf{f}) + c_2 \gamma(\mathbf{f})}{d_0 + d_1 \mathrm{TP}(\mathbf{f}) + d_2 \gamma(\mathbf{f})} \qquad (10)$$

with the constants:

$$c_0 = a_{01}\pi + a_{00} - a_{00}\pi + a_0, \quad c_1 = a_{11} - a_{10} - a_{01} + a_{00}, \quad c_2 = a_{10} - a_{00} \quad \text{and}$$
$$d_0 = b_{01}\pi + b_{00} - b_{00}\pi + b_0, \quad d_1 = b_{11} - b_{10} - b_{01} + b_{00}, \quad d_2 = b_{10} - b_{00}.$$

We assume that the joint $\mathbb{P}$ has a density $\mu$ that satisfies $d\mathbb{P} = \mu dx$, and define $\eta_m(x) = \mathbb{P}(Y_m = 1 | X = x)$. Our first main result characterizes the Bayes optimal multilabel classifier $\mathbf{f}^*_{\Psi_{\mathrm{micro}}}$.

**Theorem 2.** *Given the constants $\{c_1, c_2, c_0\}$ and $\{d_1, d_2, d_0\}$, define:*

$$\delta^* = \frac{d_2 \mathcal{U}^*_{\Psi_{micro}} - c_2}{c_1 - d_1 \mathcal{U}^*_{\Psi_{micro}}}. \qquad (11)$$

*The optimal Bayes classifier $\mathbf{f}^* := \mathbf{f}^*_{\Psi_{micro}}$ defined in (3) is given by:*

1. *When $c_1 > d_1 \mathcal{U}^*_{\Psi_{micro}}$, $\mathbf{f}^*$ takes the form $f_m^*(x) = [\![\eta_m(x) > \delta^*]\!]$, for $m \in [M]$.*

2. *When $c_1 < d_1 \mathcal{U}^*_{\Psi_{micro}}$, $\mathbf{f}^*$ takes the form $f_m^*(x) = [\![\eta_m(x) < \delta^*]\!]$, for $m \in [M]$.*

The proof is provided in Appendix A.2, and applies equivalently to instance-averaging. Theorem 2 recovers existing results in binary [9] settings (See Appendix B.1 for details), and is sufficiently general to capture many of the multilabel metrics used in practice. Our proof is closely related to the binary classification case analyzed in Theorem 2 of [9], but differs in the additional averaging across labels. A key observation from Theorem 2 is that the optimal multilabel classifier can be obtained by thresholding the marginal instance-conditional probability for each label $\mathbb{P}(Y_m = 1 | x)$ and, importantly, that the optimal classifiers for all the labels share the *same* threshold $\delta^*$. Thus, the effect of the joint distribution is only in the threshold parameter. We emphasize that while the presented results characterize the optimal population classifier, incorporating label correlations into the prediction algorithm may have other benefits with finite samples, such as statistical efficiency when there are known structural similarities between the marginal distributions [3]. Further analysis is left for future work.

The Bayes optimal for the macro-averaged population metric is straightforward to establish. We observe that the threshold is not shared in this case.

**Proposition 3.** *For a given linear-fractional metric $\Psi$, consider the macro-averaged multilabel metric $\Psi_{\mathrm{macro}}$ defined in (7). Let $c_1 > d_1 \mathcal{U}^*_{\Psi_{macro}}$ and $\mathbf{f}^* = \mathbf{f}^*_{\Psi^*_{\mathrm{macro}}}(x)$. We have, for $m = 1, 2, \ldots, M$:*

$$f_m^* = [\![\eta_m(x) > \delta_m^*]\!],$$

*where $\delta_m^* \in [0, 1]$ is a constant that depends on the metric $\Psi$ and the label-wise instance-conditional marginals of $\mathbb{P}$. Analogous results hold for $c_1 < d_1 \mathcal{U}^*_{\Psi_{macro}}$.*

**Remark 2.** *It is clear that micro-, macro- and instance- averaging are equivalent at the population level when the metric $\Psi$ is linear. This is a straightforward consequence of the observation that the corresponding sample utilities are the same. More generally, micro-, macro- and instance-averaging are equivalent whenever the optimal threshold is a constant independent of $\mathbb{P}$, such as for linear metrics, where $d_1 = d_2 = 0$ so $\delta^* = -\frac{c_2}{c_1}$ (cf. Corollary 4 of Koyejo et al. [9]). Thus, our analysis recovers known results for Hamming loss [3, 7].*

## 4    Consistent Plug-in Estimation Algorithm

Importantly, the Bayes optimal characterization points to a simple plug-in estimation algorithm that enjoys consistency as follows. First, one obtains an estimate $\hat{\eta}_m(x)$ of the marginal instance-conditional probability $\eta_m(x) = \mathbb{P}(Y_m = 1 | x)$ for each label $m$ (see Reid and Williamson [14])

using a training sample. Then, the given metric $\Psi_{\text{micro}}(\mathbf{f})$ is maximized on a validation sample. For the remainder of this manuscript, we assume wlog. that $c_1 > d_1 \mathcal{U}^*$. Note that in order to maximize over $\{\mathbf{f}_\delta : f_m(x) = [\![\eta_m(x) > \delta]\!] \ \forall m = 1, 2, \ldots, M, \delta \in (0, 1)\}$, it suffices to optimize:

$$\hat{\delta} = \underset{\delta \in (0,1)}{\operatorname{argmax}} \ \Psi_{\text{micro}}(\hat{\mathbf{f}}_\delta), \tag{12}$$

where $\Psi_{\text{micro}}$ is the micro-averaged sample metric defined in (6) (similarly for $\Psi_{\text{instance}}$). Though the threshold search is over a continuous space $\delta \in (0, 1)$ the number of distinct $\Psi_{\text{micro}}(\hat{\mathbf{f}}_\delta)$ values given a training sample of size $N$ is at most $NM$. Thus (12) can be solved efficiently on a finite sample.

---

**Algorithm 1:** Plugin-Estimator for $\Psi_{\text{micro}}$ and $\Psi_{\text{instance}}$

---

**Input**: Training examples $\mathcal{S} = \{x^{(n)}, \mathbf{y}^{(n)}\}_{n=1}^N$ and metric $\Psi_{\text{micro}}$ (or $\Psi_{\text{instance}}$).
**for** $m = 1, 2, \ldots, M$ **do**
    1. Select the training data for label $m$: $\mathcal{S}_m = \{x^{(n)}, y_m^{(n)}\}_{n=1}^N$.
    2. Split the training data $\mathcal{S}_m$ into two sets $\mathcal{S}_{m1}$ and $\mathcal{S}_{m2}$.
    3. Estimate $\hat{\eta}_m(x)$ using $\mathcal{S}_{m1}$, define $\hat{f}_m(x) = [\![\hat{\eta}_m(x) > \delta]\!]$.
**end for**
Obtain $\hat{\delta}$ by solving (12) on $\mathcal{S}_2 = \cup_{m=1}^M \mathcal{S}_{m2}$.
**Return**: $\hat{\mathbf{f}}_{\hat{\delta}}$.

---

**Consistency of the proposed algorithm.** The following theorem shows that the plug-in procedure of Algorithm 1 results in a consistent classifier.

**Theorem 4.** *Let $\Psi_{\text{micro}}$ be a linear-fractional metric. If the estimates $\hat{\eta}_m(x)$ satisfy $\hat{\eta}_m \xrightarrow{p} \eta_m$, $\forall m$, then the output multilabel classifier $\hat{\mathbf{f}}_{\hat{\delta}}$ of Algorithm 1 is consistent.*

The proof is provided in Appendix A.4. From Proposition 1, it follows that consistency holds for $\Psi_{\text{instance}}$ as well. Additionally, in light of Proposition 3, we may apply the learning algorithms proposed by [9] for binary classification independently for each label to obtain a consistent estimator for $\Psi_{\text{macro}}$.

# 5 Experiments

We present two sets of results. The first is an experimental validation on synthetic data with known ground truth probabilities. The results serve to verify our main result (Theorem 2) characterizing the Bayes optimal for averaged multilabel metrics. The second is an experimental evaluation of the plugin estimator algorithms for micro-, instance-, and macro-averaged multilabel metrics on benchmark datasets.

## 5.1 Synthetic data: Verification of Bayes optimal

We consider the micro-averaged $F_1$ metric in (9) for multilabel classification with 4 labels. We sample a set of five 2-dimensional vectors $\mathbf{x} = \{x^{(1)}, x^{(2)}, \ldots, x^{(5)}\}$ from the standard Gaussian. The conditional probability $\eta_m$ for label $m$ is modeled using a sigmoid function: $\eta_m(x) = \mathbb{P}(Y_m = 1|x) = \frac{1}{1+\exp{-w_m^T x}}$, using a vector $w_m$ sampled from the standard Gaussian. The Bayes optimal $\mathbf{f}^*(x) \in \{0, 1\}^4$ that maximizes the micro-averaged $F_1$ population utility is then obtained by exhaustive search over all possible label vectors for each instance. In Figure 1 (a)-(d), we plot the conditional probabilities (wrt. the sample index) for each label, the corresponding $f_m^*$ for each $x$, and the optimal threshold $\delta^*$ using (11). We observe that the optimal multilabel classifier indeed thresholds $\mathbb{P}(Y_m|x)$ for each label $m$, and furthermore, that the threshold is same for all the labels, as stated in Theorem 2.

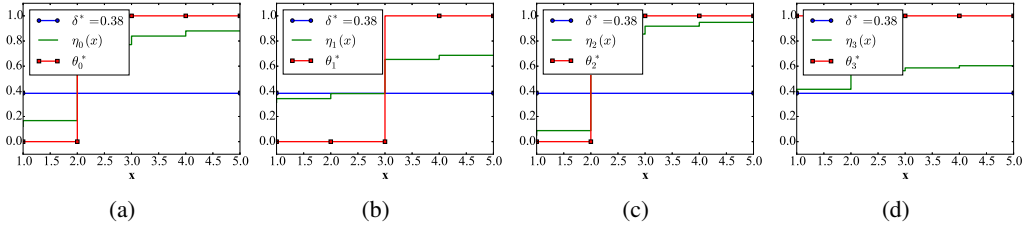

Figure 1: Bayes optimal classifier for multilabel $F_1$ measure on synthetic data with 4 labels, and distribution supported on 5 instances. Plots from left to right show the Bayes optimal classifier prediction for instances, and for labels 1 through 4. Note that the optimal $\delta^*$ at which the label-wise marginal $\eta_m(x)$ is thresholded is shared, conforming to Theorem 2 (larger plots are included in Appendix C).

## 5.2 Benchmark data: Evaluation of plug-in estimators

We now evaluate the proposed plugin-estimation (Algorithm 1) that is consistent for micro- and instance-averaged multilabel metrics. We focus on two metrics, $F_1$ and Jaccard, listed in (9). We compare Algorithm 1, designed to optimize micro-averaged (or instance-averaged) multilabel metrics to two related plugin-estimation methods: (i) a separate threshold $\delta_m^*$ tuned for each label $m$ individually – this optimizes the utility corresponding to the macro-averaged metric, but is not consistent for micro-averaged or instance-averaged metrics, and is the most common approach in practice. We refer to this as Macro-Thres, (ii) a constant threshold $1/2$ for all the labels – this is known to be optimal for averaged accuracy (equiv. Hamming loss), but not for non-decomposable $F_1$ or Jaccard metrics. We refer to this as Binary Relevance (BR) [15].

We use four benchmark multilabel datasets[4] in our experiments: (i) SCENE, an image dataset consisting of 6 labels, with 1211 training and 1196 test instances, (ii) BIRDS, an audio dataset consisting of 19 labels, with 323 training and 322 test instances, (iii) EMOTIONS, a music dataset consisting of 6 labels, with 393 training and 202 test instances, and (iv) CAL500, a music dataset consisting of 174 labels, with 400 training and 100 test instances[5]. We perform logistic regression (with $L_2$ regularization) on a separate subsample to obtain estimates of $\hat{\eta}_m(x)$ of $\mathbb{P}(Y_m = 1|x)$, for each label $m$ (as described in Section 4). All the methods we evaluate rely on obtaining a good estimator for the conditional probability. So we exclude labels that are associated with very few instances – in particular, we train and evaluate using labels associated with at least 20 instances, in each dataset, for all the methods.

In Table 1, we report the micro-averaged $F_1$ and Jaccard metrics on the test set for Algorithm 1, Macro-Thres and Binary Relevance. We observe that estimating a fixed threshold for all the labels (Algorithm 1) consistently performs better than estimating thresholds for each label (Macro-Thres) and than using threshold 1/2 for all labels (BR); this conforms to our main result in Theorem 2 and the consistency analysis of Algorithm 1 in Theorem 4. A similar trend is observed for the instance-averaged metrics computed on the test set, shown in Table 2. Proposition 1 shows that maximizing the population utilities of micro-averaged and instance-averaged metrics are equivalent; the result holds in practice as presented in Table 2. Finally, we report macro-averaged metrics computed on test set in Table 3. We observe that Macro-Thres is competitive in 3 out of 4 datasets; this conforms to Proposition 3 which shows that in the case of macro-averaged metrics, it is optimal to tune a threshold specific to each label independently. Beyond consistency, we note that by using more samples, joint threshold estimation enjoys additional statistical efficiency, while separate threshold estimation enjoys greater flexibility. This trade-off may explain why Algorithm 1 achieves the best performance in three out of four datasets in Table 3, though it is not consistent for macro-averaged metrics.

| DATASET | BR | Algorithm 1 $F_1$ | Macro-Thres | BR | Algorithm 1 Jaccard | Macro-Thres |
|---|---|---|---|---|---|---|
| SCENE | 0.6559 | **0.6847** ± 0.0072 | 0.6631 ± 0.0125 | 0.4878 | **0.5151** ± 0.0084 | 0.5010 ± 0.0122 |
| BIRDS | 0.4040 | **0.4088** ± 0.0130 | 0.2871 ± 0.0734 | 0.2495 | **0.2648** ± 0.0095 | 0.1942 ± 0.0401 |
| EMOTIONS | 0.5815 | **0.6554** ± 0.0069 | 0.6419 ± 0.0174 | 0.3982 | **0.4908** ± 0.0074 | 0.4790 ± 0.0077 |
| CAL500 | 0.3647 | **0.4891** ± 0.0035 | 0.4160 ± 0.0078 | 0.2229 | **0.3225** ± 0.0024 | 0.2608 ± 0.0056 |

Table 1: Comparison of plugin-estimator methods on multilabel $F_1$ and Jaccard metrics. Reported values correspond to *micro-averaged* metric ($F_1$ and Jaccard) computed on test data (with standard deviation, over 10 random validation sets for tuning thresholds). Algorithm 1 is consistent for micro-averaged metrics, and performs the best consistently across datasets.

| DATASET | BR | Algorithm 1 $F_1$ | Macro-Thres | BR | Algorithm 1 Jaccard | Macro-Thres |
|---|---|---|---|---|---|---|
| SCENE | 0.5695 | **0.6422** ± 0.0206 | 0.6303 ± 0.0167 | 0.5466 | **0.5976** ± 0.0177 | 0.5902 ± 0.0176 |
| BIRDS | 0.1209 | **0.1390** ± 0.0110 | **0.1390** ± 0.0259 | 0.1058 | **0.1239** ± 0.0077 | 0.1195 ± 0.0096 |
| EMOTIONS | 0.4787 | **0.6241** ± 0.0204 | 0.6156 ± 0.0170 | 0.4078 | **0.5340** ± 0.0072 | 0.5173 ± 0.0086 |
| CAL500 | 0.3632 | **0.4855** ± 0.0035 | 0.4135 ± 0.0079 | 0.2268 | **0.3252** ± 0.0024 | 0.2623 ± 0.0055 |

Table 2: Comparison of plugin-estimator methods on multilabel $F_1$ and Jaccard metrics. Reported values correspond to *instance-averaged* metric ($F_1$ and Jaccard) computed on test data (with standard deviation, over 10 random validation sets for tuning thresholds). Algorithm 1 is consistent for instance-averaged metrics, and performs the best consistently across datasets.

| DATASET | BR | Algorithm 1 $F_1$ | Macro-Thres | BR | Algorithm 1 Jaccard | Macro-Thres |
|---|---|---|---|---|---|---|
| SCENE | 0.6601 | **0.6941** ± 0.0205 | 0.6737 ± 0.0137 | 0.5046 | **0.5373** ± 0.0177 | 0.5260 ± 0.0176 |
| BIRDS | 0.3366 | **0.3448** ± 0.0110 | 0.2971 ± 0.0267 | 0.2178 | **0.2341** ± 0.0077 | 0.2051 ± 0.0215 |
| EMOTIONS | 0.5440 | **0.6450** ± 0.0204 | **0.6440** ± 0.0164 | 0.3982 | **0.4912** ± 0.0072 | **0.4900** ± 0.0133 |
| CAL500 | 0.1293 | 0.2687 ± 0.0035 | **0.3226** ± 0.0068 | 0.0880 | 0.1834 ± 0.0024 | **0.2146** ± 0.0036 |

Table 3: Comparison of plugin-estimator methods on multilabel $F_1$ and Jaccard metrics. Reported values correspond to the *macro-averaged* metric computed on test data (with standard deviation, over 10 random validation sets for tuning thresholds). Macro-Thres is consistent for macro-averaged metrics, and is competitive in three out of four datasets. Though not consistent for macro-averaged metrics, Algorithm 1 achieves the best performance in three out of four datasets.

## 6 Conclusions and Future Work

We have proposed a framework for the construction and analysis of multilabel classification metrics and corresponding population optimal classifiers. Our main result is that for a large family of averaged performance metrics, the EUM optimal multilabel classifier can be explicitly characterized by thresholding of label-wise marginal instance-conditional probabilities, with weak label dependence via a shared threshold. We have also proposed efficient and consistent estimators for maximizing such multilabel performance metrics in practice. Our results are a step forward in the direction of extending the state-of-the-art understanding of learning with respect to general metrics in binary and multiclass settings. Our work opens up many interesting research directions, including the potential for further generalization of our results beyond averaged metrics, and generalized results for DTA population optimal classification, which is currently only well-understood for the $F$-measure.

**Acknowledgments:** We acknowledge the support of NSF via CCF-1117055, CCF-1320746 and IIS-1320894, and NIH via R01 GM117594-01 as part of the Joint DMS/NIGMS Initiative to Support Research at the Interface of the Biological and Mathematical Sciences.

## Footnotes

[2]A subtle but important aspect of the definition of rank loss in the existing literature, including [2] and [7], is that the Bayes optimal is allowed to be a real-valued function and may not correspond to a label decision.

[3] Note that Hamming is typically defined as the loss, given by $1 - \Psi_{\text{Ham}}$.

[4]The datasets were obtained from `http://mulan.sourceforge.net/datasets-mlc.html`.

[5]Original CAL500 dataset does not provide splits; we split the data randomly into train and test sets.

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
