[Supplementary Material]

# A    Appendix A: Proofs

## A.1    Proof of Proposition 1

Observe that the EUM utility (2) corresponding to micro-averaged metric is simply: $\mathcal{U}(\mathbf{f}; \Psi_{\text{micro}}, \mathbb{P}) = \Psi(\text{TP}(\mathbf{f}), \text{FP}(\mathbf{f}), \text{FN}(\mathbf{f}), \text{FP}(\mathbf{f}))$ where $\text{TP}(\mathbf{f}) = \mathrm{E}_{\mathbb{P}}\left[\widehat{\text{TP}}(\mathbf{f})\right]$ ($\text{FP}(\mathbf{f}), \text{FN}(\mathbf{f}), \text{TN}(\mathbf{f})$ defined similarly). By linearity of expectation, $\mathrm{E}_{\mathbb{P}}\left[\widehat{\text{TP}}(\mathbf{f})\right] = \frac{1}{MN}\sum_{n=1}^{N}\sum_{m=1}^{M}\mathrm{E}_{\mathbb{P}}\left[\widehat{\text{TP}}(\mathbf{f})_{m,n}\right]$. But $\mathrm{E}_{\mathbb{P}}\left[\widehat{\text{TP}}(\mathbf{f})_{m,n}\right] = \mathbb{P}(f_m(x) = 1, Y_m = 1)$. Similarly, for the instance-averaged metric, we see that the corresponding $\text{TP}(\mathbf{f}) = \mathrm{E}_{\mathbb{P}}\left[\widehat{\text{TP}}(\mathbf{f})\right] = \mathbb{P}(f_m(x) = 1, Y_m = 1)$. Analogously, $\text{FP}(\mathbf{f}), \text{FN}(\mathbf{f}), \text{TN}(\mathbf{f})$ can be seen to be identical for micro- and instance-averaged metrics. Thus, whereas the sample metrics may be different, the population EUM utilities of the micro-averaged and instance-averaged metrics coincide. The second claim that $\mathbf{f}^*_{\Psi^*_{micro}} \equiv \mathbf{f}^*_{\Psi^*_{instance}}$ is immediate.

## A.2    Proof of Theorem 2

For simplicity, the proof is presented for the finite domain case. Extension to the continuous case follows directly from the approach in Theorem 2 of [9], which requires a more technical definition of the derivatives. Let $\mathcal{F} = \{\mathbf{f} : \mathcal{X} \to \mathbb{R}^M\}$, and $\mathbf{\Theta} = \{\mathbf{f} : \mathcal{X} \to \{-1, +1\}^M\} \subset \mathcal{F}$ (Note that we use the encoding $\{+1, -1\}$ for ease and it is equivalent to $\{0, 1\}$ encoding used in the main text). The derivative of $\mathcal{U}(\mathbf{f}; \Psi_{\text{micro}}, \mathbb{P})$ w.r.t. $f_m(x)$ is given by:

$$\nabla_{f_m(x)}\mathcal{U}(\mathbf{f}) = \frac{1}{(c_1 - d_1\mathcal{U}(\mathbf{f}))D_r(\mathbf{f})}\left[\eta_m(x) - \frac{d_2\mathcal{U}(\mathbf{f}) - c_2}{c_1 - d_1\mathcal{U}(\mathbf{f})}\right]\mu(x)$$

where $D_r(\mathbf{f})$ is denominator of $\mathcal{U}(\mathbf{f})$. A (multivariate) function $\mathbf{f}^* \in \mathbf{\Theta}$ optimizes $\mathcal{U}$ if $\mathbf{f}^* \in \mathbf{\Theta}$ and:

$$\sum_m \sum_{x \in \mathcal{X}} \nabla_{f_m(x)}\mathcal{U}(\mathbf{f}^*)f_m(x)dx \geq \sum_m \sum_{x \in \mathcal{X}} \nabla_{f_m(x)}\mathcal{U}(\mathbf{f}^*)f_m^*(x)dx \quad \forall\, \mathbf{f}, \mathbf{f}^* \in \mathbf{\Theta}.$$

Thus, when $c_1 \geq d_1\mathcal{U}^*$, a necessary condition for local optimality is that the sign of $f_m^*$ and the sign of $[\nabla_{f_m(x)}\mathcal{U}(\mathbf{f}^*)]$ agree point-wise wrt. $x, \forall\, m$. This is equivalent to the condition that $\text{sign}(f_m^*) = \text{sign}(\eta_m(x) - \delta^*)$. Combining this result with the constraint set $\mathbf{f} \in \mathbf{\Theta}$, we have that $f_m^* = \text{sign}(f_m^*)$, thus $f_m^* = \text{sign}(\eta_m(x) - \delta^*)$ is locally optimal. Finally, we note that $f_m^* = \text{sign}(\eta_m(x) - \delta^*)$ is unique for $\mathbf{f} \in \mathbf{\Theta}$, thus $\mathbf{f}^*$ is globally optimal. The proof for $c_1 < d_1\mathcal{U}^*$ follows using similar arguments. The observation that the threshold is shared between labels follows by definition of the gradient, where we observe that the threshold depends on the optimal utility. We note that despite the close similarity, the above result cannot be derived from Theorem 2 of [9] without significant modification to accommodate a non-iid sampling distribution i.e. different label distributions when viewed as a binary classification problem.

## A.3    Proof of Proposition 3

Note that the population utility corresponding to macro-averaged metric $\Psi_{\text{macro}}$ can be written as:

$$\mathcal{U}(\mathbf{f}; \Psi_{\text{macro}}, \mathbb{P}) = \frac{1}{M}\sum_{m=1}^{M}\Psi(\text{TP}_m(\mathbf{f}), \text{FP}_m(\mathbf{f}), \text{FN}_m(\mathbf{f}), \text{TN}_m(\mathbf{f})),$$

where $\text{TP}_m(\mathbf{f}) = \mathbb{P}(f_m(x) = 1, Y_m = 1)$ ($\text{FP}_m(\mathbf{f}), \text{FN}_m(\mathbf{f}), \text{TN}_m(\mathbf{f})$ defined similarly). Let $\mathbb{P}_m$ denote the marginal $\mathbb{P}(X, Y_m = 1)$. Note that $\mathcal{U}(\mathbf{f}; \Psi_{\text{macro}}, \mathbb{P})$ is maximized by $\mathbf{f}^*$ if, for all $m$, $\Psi(\text{TP}_m(f), \text{FP}_m(f), \text{FN}_m(f), \text{TN}_m(f))$ is optimized by $f_m^*$, where $f : \mathcal{X} \to \{0, 1\}$ and $\text{TP}_m(f) = \mathbb{P}_m(f(x) = 1, Y = 1)$. From Theorem 2 of [9], we know that the Bayes optimal $f_m^*(x)$ for linear-fractional $\Psi$ with respect to $\mathbb{P}_m$ is given by $[\![\eta_m(x) > \delta_m^*]\!]$, where $\delta_m^*$ depends only on $\Psi$ and marginal $\mathbb{P}_m$.

### A.4 Proof of Theorem 4

We first show the following claim:

**Claim.** For a fixed multilabel classifier $\mathbf{f}$, $\Psi_{\text{micro}}(\mathbf{f}; \{x^{(n)}, \mathbf{y}^{(n)}\}_{n=1}^N) \xrightarrow{P} \mathcal{U}(\mathbf{f}; \Psi_{\text{micro}}; \mathbb{P})$, where $\Psi_{\text{micro}}$ defined in (6) is the micro-averaged metric computed on iid samples.

**Proof.** The proof follows closely that of Lemma 8 of [9]. Consider the simplified empirical $\Psi_{\text{micro}}$ corresponding to the simplified population utility in (10), obtained by replacing $\text{TP}(\mathbf{f})$ with $\widehat{\text{TP}}(\mathbf{f})$, $\gamma(\mathbf{f})$ with $\hat{\gamma}(\mathbf{f}) := \frac{1}{MN} \sum_{m=1}^M \sum_{n=1}^N [\![f_m^{(n)} = 1]\!]$ and $\pi$ (in the definition of $c_0$ and $d_0$) with $\hat{\pi} = \frac{1}{MN} \sum_{m=1}^M \sum_{n=1}^N [\![y_m^{(n)} = 1]\!]$. Note that for any given $\epsilon_1 > 0, \rho > 0$, there exists $N'$ such that for any $N > N'$, $\mathbb{P}(|\widehat{\text{TP}}(\mathbf{f}) - E[\widehat{\text{TP}}(\mathbf{f})]| < \epsilon_1) > 1 - \rho/3$, $\mathbb{P}(|\hat{\gamma}(\mathbf{f}) - \gamma(\mathbf{f})| < \epsilon_1) > 1 - \rho/3$ and $\mathbb{P}(|\hat{\pi} - \pi| < \epsilon_1) > 1 - \rho/3$. Thus by union bound it follows that all the three events hold with probability $1 - \rho$. Write $c_0 = c_o' + c_0'' \pi$ and $d_0 = d_0' + d_0'' \pi$. Let $\tilde{c}_1 = 1/|c_1|$ if $c_1 \neq 0$ else let $\tilde{c}_1 = 0$. Define $\tilde{c}_2, \tilde{d}_1, \tilde{d}_2$ accordingly. Let $C = \max(\tilde{c}_1, \tilde{c}_2)$ and $D = \max(\tilde{d}_1, \tilde{d}_2)$. Note that for $\Psi_{\text{micro}}$ to be valid and bounded, $\max(C, D) > 0$. For a given $\epsilon > 0$, we want $\epsilon_1 \leq \frac{(d_0' + d_0'' \pi + d_1 \text{TP}(\mathbf{f}) + d_2 \gamma(\mathbf{f}))\epsilon}{D(\mathcal{U}(\mathbf{f}; \Psi_{\text{micro}}; \mathbb{P}) + \epsilon) + C}$, so that we can guarantee $|\Psi_{\text{micro}}(\mathbf{f}) - \mathcal{U}(\mathbf{f}; \Psi_{\text{micro}}; \mathbb{P})| < \epsilon$. Thus for all $N > N'$, for given $\epsilon$ and $\delta$, we have shown that $|\Psi_{\text{micro}}(\mathbf{f}) - \mathcal{U}(\mathbf{f}; \Psi_{\text{micro}}; \mathbb{P})| < \epsilon$ with probability at least $1 - \rho$. This completes the proof of the claim.

Assume the estimated $\hat{\eta}_m(x)$ satisfies $\hat{\eta}_m(x) \xrightarrow{P} \eta_m(x)$ as stated in the Theorem (this can be guaranteed by using a suitable class density estimation algorithm). Consider the multilabel classifier $\mathbf{f}_\delta^* = ([\![\eta_m(x) > \delta]\!])_{m=1}^M$. Let $\delta^*$ be the optimal threshold corresponding to the Bayes optimal $\mathbf{f}_{\Psi_{\text{micro}}}^*$. Because $\hat{\delta}$ is the empirical minimizer on a finite sample, we have $\Psi_{\text{micro}}(\mathbf{f}_{\hat{\delta}}^*) \geq \Psi_{\text{micro}}(\mathbf{f}_{\delta^*}^*)$ on the sample. So:

$$
\begin{aligned}
\mathcal{U}_{\Psi_{\text{micro}}}^* - \mathcal{U}(\mathbf{f}_{\hat{\delta}}^*; \Psi_{\text{micro}}, \mathbb{P}) &= \mathcal{U}_{\Psi_{\text{micro}}}^* - \Psi_{\text{micro}}(\mathbf{f}_{\hat{\delta}}^*) + \Psi_{\text{micro}}(\mathbf{f}_{\hat{\delta}}^*) - \mathcal{U}(\mathbf{f}_{\hat{\delta}}^*; \Psi_{\text{micro}}, \mathbb{P}) \\
&\leq \mathcal{U}_{\Psi_{\text{micro}}}^* - \Psi_{\text{micro}}(\mathbf{f}_{\delta^*}^*) + \Psi_{\text{micro}}(\mathbf{f}_{\hat{\delta}}^*) - \mathcal{U}(\mathbf{f}_{\hat{\delta}}^*; \Psi_{\text{micro}}, \mathbb{P}) \\
&\leq 2 \sup_\delta |\Psi_{\text{micro}}(\mathbf{f}_\delta^*) - \mathcal{U}(\mathbf{f}_\delta^*; \Psi_{\text{micro}}, \mathbb{P})| \quad\quad (13)
\end{aligned}
$$

Now, to conclude the proof, we argue that the last term in the RHS of the inequality above vanishes as $N \to \infty$. For a given $\mathbf{f} \in \mathbb{R}^M$, let $\mathcal{F}_\delta$ denote the class of multilabel classifiers obtained by thresholding $\mathbf{f}$ for some $\delta \in (0, 1)$. Using Lemma 29.1 in [6], for given $\rho$ and $\epsilon_1 > 0$, $\sup_{\mathbf{f} \in \mathcal{F}_\delta} |\widehat{\text{TP}}(\mathbf{f}) - \text{TP}(\mathbf{f})| < \epsilon_1$ with probability at least $1 - \rho/3$. Following similar arguments in the claim above, given $\epsilon > 0$ and $\rho$, we can choose $\epsilon_1$ and $N$ large enough such that $\sup_\delta |\Psi_{\text{micro}}(\mathbf{f}_\delta^*) - \mathcal{U}(\mathbf{f}_\delta^*; \Psi_{\text{micro}}, \mathbb{P})| < \epsilon$ with probability at least $1 - \rho$. This shows that the RHS of (13) vanishes as $N \to \infty$. The proof of the theorem is complete.

## B Appendix B

### B.1 Connection to Existing Results

Our results in Section 3 and Algorithm 1 generalize some of the existing results for learning with general performance metrics in binary classification. In particular, when $M = 1$, our multilabel performance metrics considered in Section 3 reduce to the linear-fractional family binary classification metrics studied by Koyejo et al. [9]. The Bayes optimal characterization in Theorem 2 of Koyejo et al. [9] can be seen as a special case of our Theorem 2. Similarly, the plugin-estimation algorithm of Koyejo et al. [9] can be derived from our Algorithm 1 for the binary case.

More recently, Narasimhan et al. [11] considered generalized performance metrics for multiclass classification and showed that the Bayes optimal (of the EUM utility) can be characterized as a thresholding of the class-conditional probability. We observe that our framework of metrics introduced in Section 2 readily gives rise to the multiclass performance metrics studied by Narasimhan et al. [11] with the additional constraint of a single label choice for each instance. While they show the Bayes optimal and consistency of learning for a different family than the linear-fractional family

we consider in this paper, many of the popular metrics including multiclass $F$-measure and multiclass Jaccard belong in both the families.

## B.2  Multilabel Decision-Theoretic Utility

Existing consistency results and algorithms for multilabel learning focus on the population utility based on decision-theoretic analysis (DTA) defined in (4). Furthermore, we are not aware of consistency results for general multilabel performance metrics. Gao and Zhou [7] define the consistency of multilabel learning with respect to DTA utility. They focus on two specific losses – Hamming and rank loss (the corresponding measures are defined in Section 2) and suggest surrogates for consistent learning with respect to the losses. Note that Hamming loss is a linear metric (i.e. it is linear in the primitives $\widehat{\mathrm{FN}}(\mathbf{f})_{m,n}$ and $\widehat{\mathrm{FP}}(\mathbf{f})_{m,n}$) and hence the Bayes optimal characterizations coincide for both DTA and EUM utility (See Equation (8) of Gao and Zhou [7]). For the Hamming loss, they showed that the Bayes optimal for the DTA utility depends on pairwise conditional distributions, i.e. $\mathbb{P}(Y_m, Y_{m'}|x)$. However, Dembczynski et al. [2] subsequently showed that the Bayes optimal for the un-normalized ranking loss indeed depends only on the label marginals $\mathbb{P}(Y_m|x)$, and in turn showed that minimizing appropriately weighted univariate loss functions (such as exponential and logistic losses) independently for $M$ labels is consistent with respect to rank loss. A subtle but important aspect of the definition of rank loss in the existing literature including [7] and [2] is that the Bayes optimal is allowed to be a real-valued function and may not necessarily correspond to an explicit label decision (note that our definition of Bayes optimal is inherently binary valued). Also, we achieve consistent learning with respect to multilabel EUM utility using a plugin-estimation algorithm. Similar plugin-estimators have been studied in the context of multilabel DTA utility maximization but only for specific metrics. [4] proposed a novel plug-in rule algorithm for estimating the parameters required for a Bayes-optimal prediction for $F$-measure (i.e., with respect to the DTA utility) via a set of multinomial regression models. Beyond theoretical analysis, further empirical study of the limits of the plug-in approach as compared to modern multilabel algorithms would be illuminating.

# C Appendix C

## C.1 Simulated Data Results

Larger plots from Figure 1 illustrating the Bayes optimal classifier for multilabel $F_1$ measure on synthetic data with 4 labels, and distribution supported on 5 instances. Plots from left to right show the Bayes optimal classifier prediction for instances, for labels 1 through 4. Note that the optimal $\delta^*$ at which the label-wise marginal $\eta_m(x)$ is thresholded is shared, conforming to Theorem 2.

(a)

(b)

(c)

(d)