[Reviews · NeurIPS 2015]

Submitted by Assigned_Reviewer_1

SUMMARY

The authors develop a theoretically justified thresholding method for plug-in estimators in multilabel classification, based on analyzing the Bayes-optimal classifier for the given objectives.

CONTRIBUTIONS: WHAT IS (ARE) THE CLAIMED CONTRIBUTION(S) OF THE PAPER? HOW SIGNIFICANT/IMPORTANT IS (ARE) THE CLAIMED CONTRIBUTION(S)?

Compute the form of the Bayes-optimal multilabel for two families of objectives, and use it to construct a multilabel classification algorithms based on plug-in estimators with good empirical performance.

The theoretical contributions add to the understanding of multilabel classification and the algorithms are a welcome finding.

SUPPORT

The theoretical results are well supported.

The algorithms are proposed in a principled way based on the theoretical results. Regarding Algorithm 1, I find the fact that the thresholds can be efficiently computed an interesting observation.

The empirical evaluation seems more illustrative than descriptive of the proposed method, but validates the theoretical findings and provides favorable evidence about their impact. Nevertheless, the empirical results presented are informative because in the theoretical results the form of the bayes-optimal classifier is derived, while in practice the class of classifiers being used is more restricted than all measurable classifiers. Section 5 therefore bridges this gap.

TECHNICAL QUALITY

The technical quality of the paper is good, but in this reviewer's opinion the text would benefit from some more discussion about certain topics, as described ahead.

There seems to be a stronger connection between the problem studied with the chosen metrics and binary classification (rather than just binary classification being a special case of the multilabel classification). Evidence to that is the seemingly strong connection between the theoretical results presented and those of Koyejo et al. 2014 (see the detailed comments section for an example). The discussion of the theoretical results would benefit from a more careful exploration of these connections.

Section 5 would be enriched by some conjectures regarding why Algorithm 1 was competitive with Macro-Thres with the macro-averaged metrics (in Table 3). It would also be interesting to see how close the thresholds of Algorithm 1 are to 1/2 where BR was competitive with the micro-averaged metrics (in Table 1).

ORIGINALITY

The paper inserts itself well in the picture, and the only suggestion I can make here is for the authors to point out that their proof techniques are substantially similar to those used by Koyejo et al. 2014 -- in particular it would be helpful for anyone checking the proofs to understand where they differ and how the treatment of multilabel classification affects the proofs.

CLARITY

The paper is clear and the text reads fluidly. There are a few missing articles in the text, but otherwise the work well written and well executed.

FOR THE REBUTTAL

If my remarks about the parallels between the authors' results and those of Koyejo et al. (2014) are pertinent, a brief comment from the authors on the topic would he useful.

DETAILED COMMENTS

I find the similarity between Theorem 2 in the paper and Theorem 2 by Koyejo et al. (2014) quite striking. Upon checking the proofs and seeing that they are essentially the same, I started to wonder the reason behind that.

I would venture that the two theorems are, in fact, equivalent. To see that (with a finite input space) we can define $\mathcal{X}' \doteq \{ x_{ij} : x_i \in \mathcal{X}, j \in [M] \}$, $P(X' = x_{ij}) \doteq \frac{1}{M}P(X = x_i)$ and $P(Y' = 1 | X' = x_{ij}) \doteq P(Y_j = 1 | X = x_i)$. Defining the quantities $\mathrm{TP}'$, $\gamma'$ and $\mathcal{L}'$ accordingly, we get

$\mathcal{L}'(f) = \frac{c_0 + c_1 \mathrm{TP}(f) + c_2 \gamma(f) }{d_0 + d_1 TP(f) + d_2 \gamma(f)}$. So we can get Theorem 2 in the paper from Theorem 2 by Koyejo et al. (2014). (Note the important influence of the decoupling over labels and the decoupling over inputs in the reduction.) A similar observation can also be made using Proposition 1 in the submission.

On the other hand the macro-averaged metric seems to be solved by combining the Bayes-optimal classifiers for $M$ separate binary classification problems.

I would also point out that (in my opinion) the threshold being shared over the labels is as impressive as its being shared over the inputs.

I was checking the proof of Theorem 2, and I arrived at a different form for the gradient $$\frac{\partial}{\partial f(x_i)_m}\mathcal{L}(f) = \frac{\mu(x_i)(c_1 - d_1 \mathcal{L}(f))}{D_r(f)}\left( \eta_m(x_i) + \frac{c_2 - d_2 \mathcal{L}(f)}{c_1 - d_1 \mathcal{L}(f)} \right)$$ (similar to the proof of Theorem 2, but with $(c_1 - d_1 \mathcal{L}(f))$ in the numerator, not in the denominator), which can also be found in Koyejo et al. (2014).

SUGGESTED FIXES

[l:36] The authors need to make sure that the citation format conforms to the conference requirements. [l:78] Remove the period [l:115] $n$ should be $N$ here (make sure you check other occurrences of $n,N$) [l:133] "corresponding... corresponds" [l:159] "metric to the confusion" [l:162] $\Psi$ was overloaded a few times up to this point, and it confused me; the usage seems consistent afterwards, but I would suggest revisiting the previous definitions and adjusting them to minimize ambiguity. [l:239] "optimizing an EUM utility" [l:244] "optimal for the macro-averaged" [l:257] "learning a multi-label" [l:351] The end of a sentence seems to be missing here. [l:406] "metrics, an optimal"

POST-REBUTTAL REMARKS

After considering that the theoretical results are a special case of previously existing results, albeit interesting observations, they are not substantial contributions, from the theoretical point of view. However, they justify the proposed algorithm, which the experiments validate and which carry interesting insights.

Summary: This is an interesting, well written paper. The theoretical contributions seem to follow from previously existing results without significant effort, but they are interesting and useful observations. The algorithm proposed is justified by these theoretical results, and validated by the empirical ones. This paper has useful theoretical observations and practical contributions for problems where metrics are based on the confusion matrix of the labels.

Submitted by Assigned_Reviewer_2

Heavy review

Summary:

The paper concerns multi-label classification. The authors analyze the problem of consistency of learning algorithms for complex performance measures. They mainly study the EUM framework in which the complex measures are computed over population-level confusion matrices. The main result states that the consistency can be achieved by a simple plug-in classifier that first estimates marginal conditional probabilities, P(y_i|x), and then tunes a common threshold for all labels. Empirical results confirm the theoretical findings.

Quality:

This is a very interesting paper. It complements recent results on consistent classifiers for complex performance measures.

The main result is not so surprising in light of recent theoretical results, however, it still had to be formally proven. Moreover, this result will have a large impact on practical applications of multi-label classification.

Minor remarks: - the paper concerns mainly the EUM framework which seems to be much easier in deriving a unified view for complex performance measures: I am looking forward for similar results for DTA.

- Jaccard and F-measure seem to have the same solution under the EUM framework on a population level (if I am not wrong) - does it also hold for the algorithmic solution, i.e., algorithms for Jaccard and F-measure deliver the same threshold?

Clarity:

The paper is clearly written.

Minor remarks: - rank loss is usually defined in terms of a scoring function: the comment given in the supplementary material should also be given in the main text as a footnote (a short version of it)

Suggestions: - for indicator function use \llbracket and \rrbracket - to be consistent with your notation you may try to use always n, m or d as iterators instead of i and j.

Originality:

The paper is original, however, it is built upon recent results obtained for binary and multi-class classification. From this perspective, the contribution is incremental, but should be sufficient for publication at NIPS.

Significance:

The presented results are significant from both theoretical and practical point of view. The introduced algorithm is quite simple, but it can serve as a baseline for more advanced approaches for MLC under complex performance measures.

After rebuttal:

Thank you for your responses.

I am rather convinced that the solution for Jaccard and F-measure is the same in the EUM framework, since Jaccard is strictly monotonic in F-measure and vice versa: F = 2J/(1+J) and J = F/(2-F).

Summary: The paper complements recent results on consistent classifiers for complex performance measures. It is very clearly written. Theoretical results are illustrated by empirical results. The paper should be considered for publication at NIPS.

Submitted by Assigned_Reviewer_3

Remark: A discussion on the statistical efficiency of estimating only one threshold (micro-avg) instead of one per label (macro-avg) would have been a plus (especially in light of Table 3 results).
Summary: A good study of plugin consistent estimators for multi-label classification. Experiments support correctly the theoretical results.

Submitted by Assigned_Reviewer_4

This paper provides an analysis of consistent multilabel classification. A general framework for defining multi-label metrics is given, and can be seen as a generalization of the Empirical Utility Maximization style classifiers that exist in the binary and multiclass settings. Then, the authors give an explicit characterization of the optimal multi-label classifier, provide a consistent plug-in estimation algorithm (with respect to the chosen multi-label metric), and perform empirical experiments that supports the theory.

The paper is very well written. The links with previous literature are clearly explained, and the original contributions are made clear. The theoretical results are completed with examples and remarks that show how to recover popular metrics using the general framework, which is appreciated for readers that are new to this kind of theoretical analysis (as myself).

The theoretical results seem novel, and also seem important as they generalize previous analyses that were made available recently in the context of binary and multiclass classification. The empirical results are interesting in the point of view of confirming that the theory works in practice.

However, did experiments comparing this method with other multi-label learning algorithms were conducted (for example, AdaBoost.MH)? Even if other approaches do not directly optimize the performance measure in hand, I think it is still interesting to have some baseline algorithms in the experiments, showing that directly optimizing the desired performance measure will indeed outperform state-of-the-art algorithms that are not designed to optimize it.

Typos and other minor comments: - The "." after "etc." in the abstract is wrongfully considered as a sentence ending by LaTeX, and a larger space is added after it. You can avoid this by using a fixed-size space: "etc.\ ". - Punctuation is sometime missing after equations (e.g. lines 110-112, 153-158). - Figure 1 is hard to read in black and white, please consider changing the line styles. - The (2) in line 351 is easy to confuse with an equation number, could be replaced by ", and (2), [...]".

- Update after rebuttal: My concerns about a comparison with baseline algorithms will be addressed in an extended version of the paper. This does not affect my (positive) review, and I keep the same evaluation score. Thanks!
Summary: The paper is well written and the theoretical results seem novel and significant. The empirical experiments verify the theory, but no comparison is made with other multi-label learning algorithms.

Author Feedback
Author rebuttal: We thank the reviewers for detailed comments. Spelling, grammar and notation suggestions have been noted and will be corrected.

Reviewer_1:
Connection to results of Koyejo et. al. 2014: Yes, the presented results for averaged binary multilabel classifiers are quite closely related to the binary classification results in Koyejo et al. 2014. As you have noted in your detailed comments, in the special case of binary averaged metrics, it is possible to convert the multi-label classification problem to a binary classification problem with respect to a certain modified distribution. Indeed, we found this connection quite interesting, and we will provide a more detailed discussion in the final version. For future work, we plan to explore if and when this intuition generalizes to additional families of multilabel classification metrics.

"how close the thresholds of Algorithm 1 are to 1/2": That is indeed a good point and matches what we observe in many cases (we will include a note in the final version).

Thank you also for delving into the proofs and making the pertinent observations. What we found significant (and you highlight) is that essentially the same proof goes through for multilabel case giving a neat interpretation that the (optimal) threshold is shared among the labels (and inputs).

Regarding the gradient, we have double checked that our calculation is correct.

Reviewer_2:
"Jaccard and F­-measure seem to have the same solution under the EUM": They have different but related thresholds (please see for instance pg. 4 of Koyejo et. al. 2014 for details)

Reviewer_3:
Comparison to other multi-label methods: The purpose of the experiments section in this paper was to illustrate the theoretical findings in practice. That said, it is worth comparing the proposed algorithms with other multi-label methods that don't optimize the specific measure. A more detailed comparison will be provided in the extended version.

Reviewer_5:
"statistical efficiency of estimating only one threshold (micro­-avg) instead of one per label": Thank you for pointing this out. We agree that this is good to emphasize. We will discuss this aspect in the final version.

Reviewer_6:
We will include error bars in the final version.

Reviewer_7:
The primarily goal of our submission is to elucidate the fundamental theoretical underpinnings of consistent multilabel classification - beyond specific choices of algorithms or parameterization. Our theoretical results show that such a general analysis is indeed possible. Further, we show that one can design simple algorithms based on the presented principled (non-heuristic) properties. The characterization of the Bayes optimal for popular multilabel metrics, including some of the metrics the reviewer has mentioned (e.g. recall, precision) have never been published, and to our knowledge were not known before the results presented in this paper. Further, the theoretical properties of various kinds of averaging have received quite limited study.

"What are the advantages of proposed measures in contrast to traditional measures?": We note that measures we analyze include traditional measures (where theoretical consistency results were unknown). To answer the larger question of why look at a large class of evaluation measures: these provide considerable flexibility for the practitioner to capture important tradeoffs e.g. for sensitive medical diagnosis. Indeed, the large class of metrics in the literature, as well as a flurry of recent results analyzing non-traditional measures, speaks to the importance of enabling this flexibility. Further, our results state that the effect of many of these modifications is simply a change to the threshold.

"Do some multi-­label learning algorithm use those proposed measures?": Most algorithms in the literature are designed to optimize hamming loss, and a few optimize F-measure. Our results show that with simple changes, similar algorithms may be designed for other losses.

"comparisons with existing multi-label learning": Our comparisons are to the current state of the art: one-vs-all multi-label classification (Macro-Thres in experiments) is in fact the most popular approach for multilabel classification, and the default approach in many packages including scikit-learn in python. It is known to be consistent for hamming loss, and empirically has been observed to be quite robust. BR (using notation from our experiments) has also been shown to have state of the art performance; see results in a recent paper [A].

[A] H.-F. Yu, P. Jain, P. Kar, and I. S. Dhillon. Large-scale Multi-label Learning with Missing Values. International Conference on Machine Learning (ICML) 31, 2014.